# Determinants of Suboptimal Gestational Weight Gain among Antenatal Women Residing in the Highest Gross Domestic Product (GDP) Region of Malaysia

**DOI:** 10.3390/nu14071436

**Published:** 2022-03-30

**Authors:** Shahrir Nurul-Farehah, Abdul Jalil Rohana, Noor Aman Hamid, Zaiton Daud, Siti Harirotul Hamrok Asis

**Affiliations:** 1Department of Community Medicine, School of Medical Sciences, Universiti Sains Malaysia, Kubang Kerian 16150, Malaysia; pare87_me@yahoo.com (S.N.-F.); na.hamid@usm.my (N.A.H.); 2Nutrition Division, Ministry of Health Malaysia, Level 1, Block E3, Complex E, Precinct 1, Putrajaya 62590, Malaysia; zaiton_daud@moh.gov.my; 3Family Health Unit, Selangor State Health Department, No 1, Wisma Sunway, Jalan Tengku Ampuan Zabedah C 9/C, Shah Alam 40100, Malaysia; ctharirotul@yahoo.com

**Keywords:** gestational weight gain, diabetes in pregnancy, pre-pregnancy BMI, maternal obesity, household income, maternal nutrition, Malaysia

## Abstract

Suboptimal gestational weight gain has been associated with adverse perinatal and maternal outcomes, including increased risk of non-communicable diseases later in life. This study aimed to determine the proportion and determinants of suboptimal GWG. A cross-sectional study was conducted among 475 pregnant women in Selangor between January and March 2020. The study included all pregnant women at their second or third trimester who fulfilled the inclusion and exclusion criteria. A multistage sampling was applied. The GWG adequacy was based on recommendations from the Institute of Medicine (2009). Multinomial logistic regressions were used for data analysis. Out of the 475 respondents, 224 (47.2%) pregnant women had inadequate GWG, 142 (29.9%) had adequate GWG, and 109 (22.9%) had excessive GWG. Multinomial logistic regression showed that having diabetes in pregnancy (AdjOR 2.24, 95% CI: 1.31, 3.83, *p* = 0.003), middle (M40) monthly household income (AdjOR 2.33, 95% CI: 1.09, 4.96, *p* = 0.029), low (B40) monthly household income (AdjOR 2.22, 95% CI: 1.07, 4.72, *p* = 0.039), and an obese pre-pregnancy BMI (AdjOR 2.77, 95% CI: 1.43, 5.35, *p* = 0.002) were significantly associated with inadequate GWG. Overweight (AdjOR 5.18, 95% CI: 2.52, 10.62, *p* < 0.001) and obese pre-pregnancy BMIs (AdjOR 17.95, 95% CI: 8.13, 36.95, *p* < 0.001) were significantly associated with excessive GWG. Improving maternal and perinatal outcomes requires targeted interventions focusing on these modifiable determinants.

## 1. Introduction

Suboptimal gestational weight gain is a major public health issue globally, affecting both mothers and offspring in the short and long term. Gestational weight gain (GWG) is defined as the amount of weight gained between the time of conception and the onset of labour, and it is contributed to by the growing foetus, amniotic fluid, placenta, uterus, plasma expansion, and mammary glands [1]. GWG is a crucial indicator of pregnant women’s nutritional status. The 2009 IOM guideline recommends the GWG based on pre-pregnancy BMI, with a higher range for underweight women and a more restricted range for overweight and obese women, i.e., 12.5–18.0 kg for underweight women, 11.5–16.0 kg for normal-weight women, 7.0–11.5 kg for overweight women, and 5.0–9.0 kg for obese women, respectively [1], with the aim of optimising perinatal and maternal outcomes in view of the growing numbers of overweight and obese women at conception.

Inadequate GWG has been associated with an increased risk of intrauterine growth restriction, low birth weight, and preterm birth [2,3], while excessive GWG has been associated with an increased risk of gestational diabetes mellitus (GDM), gestational hypertension and eclampsia, caesarean delivery, postpartum weight retention (PPWR), and macrosomic infants [3,4,5,6]. Both inadequate and excessive GWG were associated with adiposity, hypertension, and insulin resistance in offspring, independently of GDM or birth weight [7]. Hence, maternal under- and over-nutrition have been highlighted as potentially substantial contributors to the global NCD burden through their “fetal programming” impact [8]. Identifying factors of suboptimal GWG are critical for ensuring maternal and foetal health. The predictors of suboptimal GWG include age, parity, education level [9], ethnicity [10,11], household income, food security status, pre-pregnancy BMI [11,12], physical activity, dietary intake [13,14], maternal diabetes mellitus, and hypertensive disorder status [11,15].

To the best of our knowledge, there are few studies on GWG in middle-upper income countries, including Malaysia, and the findings are inconsistent [16,17]. For instance, a cross-sectional study by Chee et al. conducted in a semi-urban area in west Malaysia reported that excessive GWG was prevalent (53.3%) [16], while a study by Farhana et al. in the east-coast region of Peninsular Malaysia reported that the prevalence of inadequate GWG was higher (54.5%) [17]. Moreover, most of the published literature on the determinants of suboptimal GWG was derived from developed countries; thus, the findings might not be comparable given the differences in the sociocultural contexts. Suboptimal GWG is a key modifiable risk factor for pregnancy complications; thus, recognising these risk factors is essential for designing appropriate interventions to promote optimal GWG to improve pregnancy outcomes. It would also contribute to the evidence base for clinical practice recommendations in prenatal care in developing countries—notably, Malaysia. Managing GWG in antenatal women is critical for future child development and social adaptation. This analysis can help Malaysians. Hence, this study aims to determine the proportion of GWG categories and to identify factors associated with suboptimal GWG among pregnant women in Selangor, Malaysia.

## 2. Materials and Methods

### 2.1. Study Design and Setting

This cross-sectional study was conducted from January to March 2020 among 475 pregnant women from eighteen randomly selected health clinics in Selangor. Selangor is located in western Peninsular Malaysia and is the most populous state in the country (5.46 million), with the second highest population growth rate (2.7%) after W.P. Putrajaya (17.8%). The population, which consists of 59.3% *Bumiputera*, 27.0% Chinese, and 12.8% Indian people, best reflects the general demography of Malaysia’s population; thus, this state was selected as the best match for the Malaysian population. Selangor was the state with the third highest median monthly household income (MYR 7225) after W.P. Kuala Lumpur (MYR 7620) and W.P. Putrajaya (MYR 7512), and it was the highest GDP contributor for the nation (23.7%) [18].

### 2.2. Study Participants

The study sample consisted of all pregnant women in their second and third trimesters who fulfilled the inclusion and exclusion criteria. The inclusion criteria were being aged ≥18 years old, first trimester booking at ≤12 weeks of gestation, and being Malaysian citizens. The exclusion criteria were multiple pregnancies, absence of either height or weight records at the first trimester booking, and being unable to comprehend Malay or English.

The sample size was calculated for each variable of the determinants for suboptimal GWG using power and sample size calculation, as well as to compare two independent proportions. The largest estimated sample for each group was 475 using the proportion of adequate pregnant women with GWG with respect to the factor of Indian ethnicity (P_0_) of 0.14 [16], an estimated proportion of excessive GWG of people with Indian ethnicity (P_1_) of 0.28, 5% type 1 error, 80% power, and M (ratio between exposed and non-exposed subjects) of 1; additional 20% of missing data were applied in the data analysis. Therefore, the total sample size required was 475.

### 2.3. Data Collection and Analysis

A multistage sampling was performed. Out of nine districts and 77 health clinics in Selangor, Malaysia, four districts and eighteen health clinics were chosen using simple random sampling, followed by proportional sampling, with the total number of respondents selected from each clinic based on the average monthly attendance. Interviewer-guided questionnaires consisted of maternal and household characteristics, household food insecurity questionnaires, dietary diversity questionnaires, and pregnancy physical activity questionnaires. The Malaysian household income was categorised into three groups, namely, the top 20 percent (T20) earning ≥ MYR 9620, the middle 40 percent (M40) earning MYR 4360–9619, and the bottom 40 percent (B40) earning less than MYR 4360 per month [18]. Monthly household income was calculated as the sum of the gross monthly incomes of all household income earners. Other factors included receiving social protection [19] for any household member, anaemia (haemoglobin less than 10.5 g/dL in the second trimester or lower than 11 g/dL in the third trimester) [20], diabetes in pregnancy (both pre-existing type 2 diabetes mellitus (T2DM) and gestational diabetes mellitus) [21], and hypertension (both chronic hypertension and hypertensive disorder in pregnancy) [22]. Data on obstetric characteristics were extracted from the antenatal books. Written informed consent was obtained prior to data collection.

#### 2.3.1. Anthropometric Measurements

The pre-pregnancy weight was calculated using the first antenatal visit (before 16 weeks) minus the Institute of Medicine’s recommended weight gain in the first or second trimester [1]. The pre-pregnancy weight calculation has also been used in other studies [12]. The pre-pregnancy BMI was computed by dividing the calculated pre-pregnancy weight by the height squared, and it was categorised based on the 1995 WHO BMI cut-off point. The 1995 WHO BMI cut-off point was used due to the lack of evidence on employing the Asian BMI cut-off point when assessing GWG adequacy [23].

The GWG adequacy was determined according to the pre-pregnancy BMI status and was measured as the ratio of the actual weight gain to the expected weight gain up to the last antenatal visit by using the 2009 IOM guidelines and following the methods adapted by Laraia et al. [12]. The actual weight gain was calculated using the difference between the last antenatal visit weight and the pre-pregnancy weight. The expected weight gain was calculated as follows: expected first-trimester total weight gain + [(gestational week at last visit − 13 week) × corresponding rate of weight gain in the second and third trimesters]. The expected total weight gain in the first-trimester were 3.2, 2.2, 1.0, and 0.5 kg, while the rates corresponding to these were 0.5, 0.4, 0.3, and 0.2 kg/week for underweight, normoweight, overweight, and obese women, respectively. A ratio of less than 0.80 is regarded as a gain lower than the IOM recommendation (inadequate) and greater than 1.20 is regarded as a gain that is above the IOM recommendation (excessive), and a ratio of 0.8 to 1.2 was indicative of adequate weight gain [12].

#### 2.3.2. Household Food Security Questionnaires

This study employed a Malay version of the USDA-developed 6-item Household Food Security Survey Module (6-item HFSSM), which has been widely used internationally to assess household food insecurity among women and pregnant women [24]. The Malay version of the 6-item HFSSM was translated and validated by Mesbah et al. [25] with a Cronbach’s alpha (α) of 0.749. Affirmative responses were coded for responses of “yes”, “often”, “sometimes”, “almost every month”, and “some months but not every month”. The sum of affirmative responses referred to a household’s raw score and was used to categorise households into one of the four groups, which were food security (0 item), marginal food security (1 item), low food security (2–4 item), and very low food security (5–6 item). In this study, marginal, low, and very low food security were categorised as food insecurity [12].

#### 2.3.3. Dietary Diversity Questionnaires

The Women Dietary Diversity Score (WDDS) developed by the FAO was used to measure dietary diversity among women [26]. Dietary diversity and micronutrient adequacy are major dimensions of diet quality. The score was calculated by adding up the number of food groups consumed by each responder over a 24 h recall period. The WDDS has nine food groups formed by the aggregation of the 16 food groups that create each score. The WDDS food groups emphasised micronutrient consumption.

#### 2.3.4. Pregnancy Physical Activity Questionnaire

This study employed the Malay version of the Pregnancy Physical Activity Questionnaire (PPAQ), which was originally developed and validated by Chasan-Taber et al. [27], and the summary measures of the PPAQ were compared to the actigraph values. The validated Malay version of the PPAQ was translated by Puteri-Sulwani [28] with an intra-class correlation value ranging between 0.42 and 0.89 for all activity indices. The activities were analysed according to type, intensity, and total energy expenditure. Participants were required to report the duration of time spent on each of the activities listed in the questionnaire (household or caregiving, occupational, and sport or exercise), with the option of responding “none”, “less than half an hour per day”, “half to almost one hour per day”, “one to almost two hours per day”, “two to almost three hours per day”, or “three or more hours per day”. The intensity of each activity was determined using the metabolic equivalent (MET) values for the activities from the Compendium of Physical Activities. The mean energy expenditure per week (MET-hour/week) was calculated for each activity by multiplying the duration of the activity by its intensity. The total physical activity was computed by summing the MET-hours/week of each activity.

Data were analysed using IBM SPSS Statistics 24 (SPSS Inc., Chicago, IL, USA). The proportions of GWG categories and participants’ characteristics were described using descriptive statistics with the mean and standard deviation (SD) or median and interquartile range (IQR) based on the normality distribution for numerical variables or frequencies and percentages for categorical variables. Simple and multinomial logistic regression was performed to identify factors associated with GWG. The dependent variable categories were coded as 0, 1, and 2; adequate GWG was coded as 0 as a reference group, while the codes of 1 and 2 denoted inadequate and excessive GWG, respectively. Multicollinearity between different predictor variables was checked using the variance inflation factor (VIF). A value of less than 5.0 indicated that there were no multicollinearity issues. All possible two-way interaction terms between significant variables were checked one at a time. A *p*-value < 0.05 was regarded as statistically significant.

## 3. Results

### 3.1. Proportions of GWG Categories

Of the 475 respondents, the proportions of inadequate, adequate, and excessive GWG in this study were 47.2% (*n* = 224), 29.9% (*n* = 142), and 22.9% (*n* = 109).

### 3.2. Characteristics of the Pregnant Women

The participants’ mean age was 30.17 ± 4.69, with 79.4% aged below 35 years old. Most of the participants were Malays (87.4%), had lower secondary education (40.6%), and were employed (59.8%). The majority of spouses had secondary education (51.6%), and all were employed. Nearly half of the participants’ households (46.5%) earned between MYR 4360 and 9619 monthly. More than half of them did not own a house (61.9%), were food secure (70.9%), and spent ≥ MYR 500 monthly for food (50.5%). The mean dietary diversity score was 4.74 ± 1.28. None of the respondents consumed alcohol or smoked. Most of the women were multiparous (55.6%) and had pre-a pregnancy BMI that was overweight or obese (53.0%). The majority received nutritional counselling (82.9%). Only 28.2% had diabetes mellitus, 3.6% had hypertension, and 39.6% had anaemia in pregnancy. The characteristics of pregnant women according to the GWG categories are summarised in Table 1.

### 3.3. Physical Activity of Pregnant Women

Overall, the median (IQR) of the total energy expenditure of the pregnant women was 148.26 (131.56) MET-hour/week, with the excessive GWG group recording the highest median energy expenditure of 152.19 (152.11). For the intensity of physical activity (PA), the highest median (IQR) was for light PA, followed by moderate PA and sedentary PA with medians (IQR) of 104.75 (91.85), 28.86 (53.07), and 7.35 (11.82) MET-hr/day, respectively; there was no median for vigorous activity. In terms of the type of physical activity, the highest median (IQR) was for household or caregiving activity at 73.15 (82.26) MET-hr/day, followed by inactivity, transportation, and sports or exercise at 10.72 (15.88), 10.71 (21.28), and 1.28 (4.22) MET-hr/day, respectively (Table 2).

### 3.4. Determinants of GWG

Simple multinomial logistic regression showed that the maternal and spouse’s education level, maternal employment status, monthly household income, house ownership, receiving of social protection, food security status, monthly food expenditure, pre-pregnancy BMI, antenatal nutritional counselling, diabetes mellitus, and anaemia status were associated with suboptimal GWG with a *p*-value < 0.25. In the final multinomial logistic regression model, pregnant women who were overweight and obese pre-pregnancy had odds of having excessive GWG that were 5.18 and 17.95 times higher compared to women with a normal pre-pregnancy weight. However, women who were obese pre-pregnancy (AdjOR: 2.77, 95% CI: 1.43, 5.35), who were in the B40 (AdjOR: 2.22, 95% CI: 1.07, 4.72) and M40 (AdjOR: 2.33, 95% CI: 1.09, 4.96) monthly household income groups, and who had diabetes during pregnancy (AdjOR: 2.24, 95% CI: 1.31, 3.83) were significantly associated with inadequate GWG (Table 3).

## 4. Discussion

### 4.1. Proportion of Suboptimal GWG

Globally, less than half of pregnant women gain weight within the recommended range for GWG. A meta-analysis of over one million women across continents and different ethnicities found that only 24.5% of pregnant women gained weight within the IOM recommendations [4]. The present study found that less than a third of pregnant women had adequate GWG. Our findings were consistent with those of a recent study that documented the significant burden of inadequate GWG in most low- and middle-income countries [29]. In contrast, in high-income countries, such as the USA and those in Western Europe, the prevalence of excessive GWG was far higher [4]. In comparison to local studies, this study is consistent with previous studies that reported suboptimal GWG ranges between 57.8 and 82.2% [5,16]. In addition, forms of malnutrition, undernutrition, and overnutrition existed in this country. For instance, Chee et al. [16] and Farhana et al. [30] found a higher prevalence of excessive GWG among pregnant women in urban areas in Johor and Kelantan, respectively, while Farhana et al. [17] and the current study found a higher prevalence of inadequate GWG among pregnant women in rural areas in Gua Musang and Kelantan and urban areas in Selangor. The double burden of malnutrition in developing countries was attributed to socioeconomic developments and urbanisation, leading to a transition of nutrition with increasingly unhealthy dietary patterns and sedentary lifestyles. This contributed to the rising trend of overweight and obesity amidst the prevailing undernutrition and micronutrient deficiency in Malaysia. Aside from that, the study population and pre-pregnancy BMI calculation technique method influenced the reported results. For instance, Chee et al. [16] employed self-reported pre-pregnancy BMI, while Wang et al. [31] used the antenatal booking weight to calculate BMI. Lin et al. reported that obese women were more likely to under-report their weight by ≥5% compared to normal-weight women, whereas underweight women were more likely to over-report their weight [32]. This might affect computations of GWG, in which the GWG for obese and overweight women will be overestimated, whereas the GWG for underweight women will be underestimated, leading to misclassification. In addition, using the BMI from the first booking as a proxy for pre-pregnancy BMI may lead to inaccurate assessment of the adequacy of GWG, as, on average, pregnant women gain 1–2 kg during the first trimester [1]. In this study, 1.26% of the respondents’ pre-pregnancy BMI values were misclassified if early-booking BMI was used without adjustment in the analysis of the adequacy of GWG.

### 4.2. Determinants of Suboptimal GWG

Our study showed that both pre-pregnancy overweight and obesity status were independent risk factors for suboptimal GWG. Specifically, pre-pregnancy BMI ≥ 29.0 kg/m^2^ increased the risk of both inadequate and excessive GWG. In fact, in this study, less than a third of obese pregnant women had the recommended GWG. A similar finding was reported in other studies [11]. The observed findings could be explained by the more restrictive recommended GWG range for overweight and obese women, thus making them more likely to gain excessively, although obese women gain less on average than underweight or normal-weight women (mean of 4 kg vs. 9 kg) [33]. This has significant implications considering that the number of overweight and obese pregnant women in upper- and middle-income countries has sharply risen, including in Malaysia [34,35]. Hence, more pregnant women will be at risk of suboptimal GWG. Neonates of overweight and obese women who gained less than the recommended weight (<5 kg) or lost weight during pregnancy are at risk of SGA and have lower fat, lean body mass, smaller length and head circumference, low birth weight, and premature delivery [1,15]. Moreover, excessive GWG increases the risk of postpartum weight retention and subsequent obesity in women of reproductive age and their offspring [1].

The present study found that after adjusting for pre-pregnancy BMI and monthly household income, pregnant women with diabetes had double the risk of inadequate GWG. Consistently with this finding, studies have shown that overweight and obese women who gained <5 kg had a greater proportion of diabetes or abnormal OGTT than those who gained >5 kg (*p* = 0.002) [15]. Moreover, a recent cohort study of 2842 diabetic pregnant women found that inadequate GWG was prevalent, with the majority (50.3%) having inadequate GWG, followed by adequate (31.6%) and excessive GWG (18.1%) [6]. Hui et al. reported that fear of insulin therapy resulted in obsessive carbohydrate restriction among diabetic pregnant women, as achieving the blood glucose target for foetal wellbeing outweighed the concerns about unbalanced meals [36]. The risk of ketogenesis, particularly in overt hyperglycaemia and/or weight loss, is associated with neurocognitive outcomes in offspring who are born to mothers with pre-existing diabetes or GDM [37].

This study found that lower- and middle-income households were more likely to have inadequate GWG compared to their wealthier counterparts. This finding is consistent with those of other studies that reported a significant association between low household income and inadequate GWG [9]. In Malaysia, income classifications aid the government in adequately planning social security schemes and in the formulation of national development plans. Although the median household income has increased more than six-fold after the country’s GDP since 1970 [38], approximately 1.6 million urban residents were classified as B40, with the highest concentration in Selangor’s urban areas (16.6 %) [39]. The B40 group is constantly confronted with rising costs of living, particularly in urban areas, given that their monthly income is insufficient to meet basic needs, as the majority of the B40 population consists of low-skilled and casual workers who are vulnerable to job insecurity and financial hardship [40]. According to Son and Ismail, consumption in the bottom 20% of households was confined to ‘basic needs’, with food, shelter, and clothing being the primary products consumed [41]. In this study, a higher proportion of pregnant women with food insecurity (56.5%), a housewife (55.4%), or women who stopped working during pregnancy (47.1%), as well as those in households receiving social protection (51.8%), had inadequate GWG (row percentage). Moreover, pregnant women with suboptimal GWG had lower mean dietary variety scores than those with adequate GWG. Consistently with this finding, the Khazanah Research Institute (KRI), in its recent study “Are both the B40 and M40 ‘poor’?”, reported that a significant proportion of the M40 group in the country had consumption patterns comparable to those of the B40 group’s upper echelons [41]. In other words, economic hardship also affected the middle class (M40), a classic “urban poor” struggle. Thus, the income classifications were recently revised to reflect rising living costs, inflation, household size, and location [38]. When income is restricted, households may be forced to prioritise necessities above food budgets. Edin and Lein [42] reported that poor urban mothers choose to go without food rather than other necessities, such as medical care. Several local studies found no significant associations between income and GWG due to different income cut-off points [16,30].

With regards to physical activity (PA), by intensity, light PA had the highest intensity, and no vigorous activity was reported, while household and caregiving activities were the most prevalent types of PA performed during pregnancy. Similarly, a study in the upper-middle income country of the Dominican Republic reported that household and caregiving PA accounted for the majority of energy expenditure (56.0%), whereas sports and exercise accounted for the least (6.4%) [43]. Yong et al. also found that women with high sedentary behaviour had a greater risk of GDM despite their high PA [13]. This study found no significant association of maternal age, ethnicity, household size, number of children under care, dietary diversity score, total activity, parity, and hypertension status with suboptimal GWG. A careful interpretation of the findings is needed due to the small sample size representing other ethnicities, the number of children under care that is greater than four, and hypertensive status.

### 4.3. Strengths

To the best of our knowledge, the present study is the first population-based study in Malaysia to show the association of pre-pregnancy BMI, household income, and diabetes in pregnancy status with suboptimal GWG. While most published studies highlighted one dimension of GWG, which was excessive [5,16,30], this study looked at both forms of malnutrition—inadequate and excessive GWG.

### 4.4. Limitations

This study is not without limitations. First, the survey was conducted at one specific point in time. Thus, the factors in this study may change during the course of pregnancy, which may influence the adequacy of GWG. In addition, the data collection on 160 (24.4%) participants was extended to the first week of the movement control order (MCO) in Malaysia due to COVID-19 [40], which potentially affected the study’s results. However, the GWG was measured across time rather than at a single point in time, as described in the method section, thus mitigating the MCO’s impact. Second, the study’s sample population was largely middle- and low-income women, reflecting the population that used the government-funded free antenatal care services, whereas the majority of high-income women obtained prenatal care from a private health centre [44]. Hence, the generalisability of the findings should be taken with caution. Thirdly, though adjustment was performed for several potential confounders, residual confounding may still exist. Nevertheless, rather than employing individual binary logistic regression, we used a multinomial logistic regression model to overcome the limitations of separate binary models, such as redundancy, information loss from only analysing a subset of data at once, and repeated analysis issues from analysing several pairs of categories, thus providing more precise parameter estimates from simultaneous estimations [45]. Lastly, further studies are recommended in order to explore social phenomena particular to pregnant women with suboptimal GWG in order to gain in-depth understanding of the determinants of suboptimal GWG and develop interventions to promote adequate GWG.

## 5. Conclusions

In conclusion, the present study suggests that more than two-thirds of pregnant women have suboptimal GWG, with the highest proportion being in the inadequate GWG group. The determinants of suboptimal GWG in this study were overweight and obese pre-pregnancy BMI, having diabetes mellitus, and low and middle monthly household income. More importantly, these determinants are modifiable. To ensure that Malaysia achieves SDG 3 by 2030 in order to reduce maternal and under-five mortality and reduce the intergenerational effect on non-communicable diseases (NCDs) of suboptimal GWG, interventions should address both arms of GWG—under- and over-nutrition. In addition, a sustainable multi-agency strategy is essential for ensuring income protection for pregnant women who are at risk during the vital phase of foetal development.

## Figures and Tables

**Table 1 nutrients-14-01436-t001:** Characteristics of the pregnant women based on the gestational weight gain categories (*n* = 475).

Variables	Gestational Weight Gain (*n*, %)			
	Total (*n* = 475) Mean (SD), *n* (%)	Inadequate (*n* = 224) Mean (SD), *n* (%)	Adequate (*n* = 142) Mean (SD), *n* (%)	Excessive (*n* = 109) Mean (SD), *n* (%)
Age (years)	30.2 (4.7)	30.2 (4.7)	29.8 (4.8)	30.5 (4.5)
Ethnicity				
Malay	415 (87.4)	193 (86.2)	125 (88.0)	97 (89.0)
Chinese	22 (4.6)	11 (4.9)	6 (4.3)	5 (4.6)
Indian	32 (6.7)	17 (7.6)	9 (6.3)	6 (5.5)
Others	6 (1.3)	3 (1.3)	2 (1.4)	1 (0.9)
Maternal education level				
Primary or less	12 (2.5)	8 (3.5)	2 (1.4)	2 (1.8)
Secondary	193 (40.6)	99 (44.2)	53 (37.3)	41 (37.6)
Pre-university	145 (30.5)	68 (30.4)	49 (34.5)	28 (25.7)
Tertiary	125 (26.3)	49 (21.9)	38 (26.8)	38 (34.9)
Maternal employment status				
Unemployed	157 (33.1)	87 (38.8)	42 (29.6)	28 (25.7)
Employed	284 (59.8)	121 (54.0)	91 (64.1)	72 (66.1)
Stopped working due to pregnancy	34 (7.1)	16 (7.1)	9 (6.3)	9 (8.3)
Household characteristics				
Husband’s education level	16 (3.4)	8 (3.6)	5 (3.5)	3 (2.8)
Primary or less	245 (51.6)	129 (57.6)	59 (41.5)	57 (52.3)
Secondary	134 (28.2)	52 (23.2)	53 (37.3)	29 (26.6)
Pre-university	80 (16.8)	35 (15.6)	25 (17.6)	20 (18.3)
Monthly household income				
Low (B40 group)	204 (43.0)	105 (46.9)	62 (43.7)	37 (33.9)
Middle (M40 group)	221 (46.5)	104 (46.4)	59 (41.5)	58 (53.2)
High (T20 group)	50 (10.5)	15 (6.7)	21 (14.8)	14 (12.8)
House ownership status				
No	294 (61.9)	143 (63.8)	93 (65.5)	58 (53.2)
Yes	181 (38.1)	81 (36.2)	49 (34.5)	51 (46.8)
Household size				
1–4	281 (59.2)	128 (57.1)	87 (61.3)	66 (60.6)
>4	194 (40.8)	96 (42.9)	55 (38.7)	43 (39.4)
No. of children under care				
1–4	469 (98.7)	222 (99.1)	139 (97.9)	108 (99.1)
>4	6 (1.3)	2 (0.9)	3 (2.1)	1 (0.9)
Recipient of social protection program				
No	334 (70.3)	151 (67.4)	107 (75.4)	76 (69.7)
Yes	141 (29.7)	73 (32.6)	35 (24.6)	33 (30.3)
Household food security status				
Secure	337 (70.9)	146 (65.2)	110 (77.5)	81 (74.3)
Insecure	138 (29.1)	78 (34.8)	32 (22.5)	28 (25.7)
Monthly food expenditure (MYR) ^1^				
<350	161 (33.9)	71 (31.7)	51 (35.9)	39 (35.8)
350–499	74 (15.6)	36 (16.1)	16 (11.3)	22 (20.2)
≥500	240 (50.5)	117 (52.2)	75 (52.8)	48 (44.0)
Women dietary diversity score	4.74 (1.28)	4.71 (1.36)	4.82 (1.25)	4.72 (1.15)
Obstetric characteristics				
Pre-pregnancy weight (kg)	64.84 (16.38)	64.95 (17.02)	58.19 (13.39)	73.27 (14.69)
Height (m)	1.56 (0.06)	1.56 (0.06)	1.57 (0.06)	1.57 (0.06)
Pre-pregnancy BMI (kg/m^2^)				
Underweight	33 (6.9)	16 (7.1)	16 (11.3)	1 (0.9)
Normal weight	190 (40.0)	93 (41.5)	79 (55.6)	18 (16.5)
Overweight	116 (24.4)	50 (22.3)	32 (22.5)	34 (31.2)
Obese	136 (28.6)	65 (29.0)	15 (10.6)	56 (51.4)
Gravida				
Primigravida (G1)	155 (32.6)	64 (28.6)	51 (35.9)	40 (36.7)
Multigravida (G2–G4)	264 (55.6)	130 (58.0)	80 (56.3)	54 (49.5)
Grandmultipara (G5>)	56 (11.8)	30 (13.4)	11 (7.7)	15 (13.8)
Nutritional counselling				
No	81 (17.1)	39 (17.4)	17 (12.0)	25 (22.9)
Yes	394 (82.9)	185 (82.6)	125 (88.0)	84 (77.1)
Comorbidities				
Diabetes in pregnancy				
Pre-existing diabetes mellitus	13 (2.7)	10 (4.5)	1 (0.7)	2 (1.8)
Gestational diabetes mellitus	121 (25.5)	73 (32.6)	24 (16.9)	24 (22.0)
No diabetes mellitus	341 (71.8)	141 (62.9)	117 (82.4)	83 (76.1)
Hypertension				
Pre-existing hypertension	11 (2.3)	5 (2.2)	3 (2.1)	3 (2.8)
Hypertensive disorder in pregnancy	6 (1.3)	4 (1.8)	1 (0.7)	1 (0.9)
No hypertension	458 (96.4)	215 (96.0)	138 (97.2)	105 (96.3)
Anaemia				
No	287 (60.4)	143 (63.8)	69 (48.6)	75 (68.8)
Yes	188 (39.6)	81 (36.2)	73 (51.4)	34 (31.2)

^1^ One USD was equivalent to MYR 4.18 as of 17 February 2022.

**Table 2 nutrients-14-01436-t002:** Physical activity during pregnancy according to the GWG categories (*n* = 475).

Physical Activity	Total Median (IQR)	Gestational Weight Gain Median (IQR)		
		Inadequate (*n* = 224)	Adequate (*n* = 142)	Excessive (*n* = 109)
Total energy expenditure (MET-hr/day)	148.26 (131.56)	147.42 (136.55)	148.23 (116.91)	152.19 (152.11)
By intensity				
Sedentary	7.35 (11.82)	7.35 (13.43)	7.35 (10.73)	7.35 (10.50)
Light	104.75 (91.85)	100.13 (92.70)	106.89 (97.79)	109.76 (85.53)
Moderate	28.86 (53.07)	27.97 (49.26)	29.48 (55.20)	33.53 (73.76)
Vigorous	0.00 (0.00)	0.00 (0.00)	0.00 (0.00)	0.00 (0.00)
By type				
Household/caregiving	73.15 (82.26)	74.40 (89.43)	68.69 (76.76)	80.74 (78.09)
Sports/exercise	1.28 (4.22)	1.60 (4.22)	1.08 (4.22)	1.60 (4.50)
Transportation	10.71 (21.28)	10.71 (21.28)	10.71 (19.74)	15.96 (26.53)
Inactivity	10.72 (15.88)	13.65 (15.79)	7.35 (13.43)	10.72 (24.28)

**Table 3 nutrients-14-01436-t003:** Final multinomial logistic regression of factors associated with suboptimal GWG (*n* = 475).

Variables	Inadequate GWG (*n* = 224)			Excessive GWG (*n* = 109)				
	Regression Coefficient (b)	AdjOR (95% CI)	Wald Statistic (df)	*p*- Value	Regression Coefficient (b)	AdjOR (95%CI)	Wald Statistic (df)	*p*- Value
Diabetes during pregnancy								
No		1.00				1.00		
Yes	0.81 (0.27)	2.24 (1.31,3.83)	8.72 (1)	0.003	−0.29 (0.35)	0.75 (0.38,1.48)	0.70 (1)	0.403
Monthly household income								
High (T20 group)		1.00				1.00		
Middle (M40 group)	0.84 (0.39)	2.33 (1.09,4.96)	4.78 (1)	0.029	0.14 (0.44)	1.14 (0.49,2.68)	0.10 (1)	0.755
Low (B40 group)	0.80 (0.39)	2.22 (1.07,4.72)	4.25 (1)	0.039	−0.43 (0.45)	0.65 (0.27,1.55)	0.90 (1)	0.343
Pre-pregnancy BMI								
Normal weight		1.00				1.00		
Underweight	−0.23 (0.39)	0.80 (0.37,1.72)	0.34 (1)	0.796	−1.25 (1.07)	0.29 (0.04,2.33)	1.36 (1)	0.243
Overweight	0.12 (0.28)	1.12 (0.64,1.95)	0.17 (1)	0.685	1.64 (0.37)	5.18 (2.52,10.62)	20.12 (1)	<0.001
Obese	1.02 (0.34)	2.77 (1.43,5.35)	9.15 (1)	0.002	2.89 (0.40)	17.95 (8.13,39.65)	51.03 (1)	<0.001

Variable selection was applied with the Enter method. No multicollinearity and no interaction. The classification table was 51.2% correctly classified. The area under the receiver operating characteristic (ROC) curve was 66.0% for inadequate GWG. The area under the receiver operating characteristic (ROC) curve was 81.0% for excessive GWG.

## Data Availability

The data presented in this study are available on request from the corresponding author. The data are not publicly available due to privacy restrictions.

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
