# Peer review of "Determinants of Suboptimal Gestational Weight Gain among Antenatal Women Residing in the Highest Gross Domestic Product (GDP) Region of Malaysia"

_nutrients, 2022, doi:10.3390/nu14071436_

Round 1

Reviewer 1 Report

To control of gestational weight gain among antenatal women is very important for developing children and social adaption in the future. This analysis can help for Malaysian.

Author Response

Point 1: To control of gestational weight gain among antenatal women is very important for developing children and social adaption in the future. This analysis can help for Malaysian..

Response 1: The statement has been added in the manusript

Reviewer 2 Report

This is well-written paper there are a few suggestions below.

Major. The finding with GWG seems abit at odd with what one would expcct.

The first item in the results should be the women’s pre-pregnancy BMI and whether they gained too little just right or too much weigh based on IOM guidelines. The percent are not presented correctly. What is important is the number of women in each category that gained the right amount of weight.

Underweight women number 33. So inadequate, adequate, and excessive, should be 49%, 33%, and 3% respectively. Obese is 136 and the corresponding numbers should be 47, 11, 42, respectively. Overweight is 116. 43%, 28%, 29. These are all pretty much the same

Minor: I think this journal uses American spellings, the paper uses British

None of the acronyms are defined in the abstract.

“The cut-off values of GWG adequacy” do you mean cutoff values derived from the IOM, which are based on pre-pregnancy BMI, were used.

Avoid using B40 and M40 in the abstract.

Introduction

Gestational weight gain (GWG) is defined as the amount of weight gained between the time of conception and the onset of labour, which is contributed to by the growing foetus, amniotic fluid, placenta, uterus, plasma expansion, and mammary gland

“GWG is a crucial indicator of pregnant women’s nutritional status. In 2009, the Institute of Medicine….” Can you shorten this just to say that weight gain recommendations during pregnancy are based on pre-pregnancy BMI with women with lower BMIs advised to gain more than those with higher BMIs.

“Inadequate” suggest that weight gain was not it is really inappropriate.

Hence, maternal under-and over-nutrition have been highlighted as potentially substantial contributor to the global NCD burden through their "foetal program

ming" impact [8]. Identifying factors of suboptimal GWG is critical to ensuring maternal and foetal health. Among the predictors of suboptimal GWG include age, parity, education level [9], ethnicity [10, 11], household income, food security status and pre-pregnancy BMI [11, 12]; physical activity and dietary intake [13, 14]; maternal diabetes mellitus and hypertensive disorder status [11, 15].

Suggest condensing into modifiable risk factors

To the best of our knowledge, there are limited published studies on GWG in developing countries including Malaysia, and the findings are inconsistent [16, 17].

There are few studies on GWG in developing countries including Malaysia, and the findings are inconsistent [16, 17].

I don’t consider Malaysia a developing a country. It is an upper middle-income country. There is no doubt there are poor places in Malaysia, but Salangor?

The study is presented as a representative survey, yet we have no data to judge this. Did the sampling frame include those receiving private care? How many women refused participation? How closely dose the sample match the actual population.

Have these IOM cutoffs been validated in Malaysian women.

Author Response

Response to Reviewer 1 Comments

Point 1: Major: The first item in the results should be the women’s pre-pregnancy BMI and whether they gained too little just right or too much weigh based on IOM guidelines. The percent are not presented correctly. What is important is the number of women in each category that gained the right amount of weight. Underweight women number 33. So inadequate, adequate, and excessive, should be 49%, 33%, and 3% respectively. Obese is 136 and the corresponding numbers should be 47, 11, 42, respectively. Overweight is 116. 43%, 28%, 29. These are all pretty much the same

Response 1: The aim of this study is to identify the proportion and factors associated with suboptimal GWG. Hence, we do not specifically present the results with regards to GWG outcomes based on pre-pregnancy BMI. The percentage that was presented in Table 1 was a column percentage, rather than a row percentage. 

Point 2: Minor: I think this journal uses American spellings, the paper uses British

Response 2: The words spelled in American has been changed to British 

Point 3: None of the acronyms are defined in the abstract.

Response 3: All of the acronyms in abstract has been removed, replaced with complete sentences.

Point 4: “The cut-off values of GWG adequacy” do you mean cutoff values derived from the IOM, which are based on pre-pregnancy BMI, were used.

Response 4: The ‘cut-off’ has been removed

Point 5: Avoid using B40 and M40 in the abstract.

Response 5: We believe it is crucial to emphasise the B40 and M40 groups in the abstract in order to enlighten policymakers, given that this term is unique to the population in Malaysia that was utilised to construct social protection programs.

Point 6: Introduction

Gestational weight gain (GWG) is defined as the amount of weight gained between the time of conception and the onset of labour, which is contributed to by the growing foetus, amniotic fluid, placenta, uterus, plasma expansion, and mammary gland

Response 6: We believe that it is critical for us to explain what the GWG is and how it came to be, that is, the GWG's fundamental notion, to be introduced early in the introduction of GWG so that the reader has a better understanding of GWG. 

Point 7:“GWG is a crucial indicator of pregnant women’s nutritional status. In 2009, the Institute of Medicine….” Can you shorten this just to say that weight gain recommendations during pregnancy are based on pre-pregnancy BMI with women with lower BMIs advised to gain more than those with higher BMIs.

Response 7: The sentence has been revised according to the suggestion

Point 8: Hence, maternal under-and over-nutrition have been highlighted as potentially substantial contributor to the global NCD burden through their "foetal programming" impact [8].

Response 8: "Foetal programming" is a specific term that refers to the in-utero insult occurring during critical points of pregnancy that predisposes disease to adulthood (Barker, 2003). Hence, we believe it is crucial to retain “foetal programming” in the above sentence.

Point 9: Identifying factors of suboptimal GWG is critical to ensuring maternal and foetal health. Among the predictors of suboptimal GWG include age, parity, education level [9], ethnicity [10, 11], household income, food security status and pre-pregnancy BMI [11, 12]; physical activity and dietary intake [13, 14]; maternal diabetes mellitus and hypertensive disorder status [11, 15].

Suggest condensing into modifiable risk factors

Response 9: The factors provided included both modifiable (physical activity, diet) and non-modifiable (age, ethnicity) variables that have a significant association evidences with the outcome of the GWG; thus, we believe it is critical to state those variables.

Point 10: To the best of our knowledge, there are limited published studies on GWG in developing countries including Malaysia, and the findings are inconsistent [16, 17].

There are few studies on GWG in developing countries including Malaysia, and the findings are inconsistent [16, 17].

I don’t consider Malaysia a developing a country. It is an upper middle-income country. There is no doubt there are poor places in Malaysia, but Salangor?

Response 10: ‘developing countries’ has been changed to ‘middle income countries’

Point 11: The study is presented as a representative survey, yet we have no data to judge this. Did the sampling frame include those receiving private care? How many women refused participation? How closely dose the sample match the actual population.

Response 11: This study employed probability sampling, with a multistage sampling approach, where simple random sampling used to select districts and health clinics, followed by proportionate sampling. Additionally, Selangor, the state chosen for this study, had an ethnic composition that was nearly identical to that of the Malaysian population as has been stated in method section. Nevertheless, we were aware the sample population were not covering those who received care in the private health care and was stated under the limitation.

Point 12: Have these IOM cutoffs been validated in Malaysian women.

Response 12: The evidence for IOM cut-off points for multiethnic groups was limited, given that Asian groups have distinct BMI cut-off points, whereas the IOM weight increase recommendations are based on Western WHO BMI cut-offs (Arora and Tamber Aeri, 2019; Rasmussen and Yaktine, 2009). Until new evidence emerged, the IOM guideline recommended these recommendations be applied to all women regardless of ethnicity.

Arora, P. & Tamber Aeri, B. (2019). Gestational Weight Gain among Healthy Pregnant Women from Asia in Comparison with Institute of Medicine (IOM) Guidelines-2009: A Systematic Review. Journal of Pregnancy, 2019, 3849596. doi: 10.1155/2019/3849596

Barker, D. J. P. (2003). Editorial: The Developmental Origins of Adult Disease. European Journal of Epidemiology, 18(8), 733-736.

Rasmussen, K. & Yaktine, A. (2009). Institute of Medicine (US). Committee to Reexamine IOM Pregnancy Weight Guidelines. Weight gain during pregnancy: reexamining the guidelines. Retrieved from:https://www.ncbi.nlm.nih.gov/books/NBK32815/ [Accessed 08 March 2021].
